# Effect of the Solvate Environment of Lithium Cations on the Resistance of the Polymer Electrolyte/Electrode Interface in a Solid-State Lithium Battery

**DOI:** 10.3390/membranes12111111

**Published:** 2022-11-08

**Authors:** Alexander V. Chernyak, Nikita A. Slesarenko, Anna A. Slesarenko, Guzaliya R. Baymuratova, Galiya Z. Tulibaeva, Alena V. Yudina, Vitaly I. Volkov, Alexander F. Shestakov, Olga V. Yarmolenko

**Affiliations:** 1Federal Research Center of Problems of Chemical Physics and Medicinal Chemistry RAS, 142432 Chernogolovka, Russia; 2Scientific Center in Chernogolovka RAS, 142432 Chernogolovka, Russia; 3Faculty of Fundamental Physical and Chemical Engineering, M. V. Lomonosov Moscow State University, 119991 Moscow, Russia

**Keywords:** polymer electrolyte, nanocomposite, organic electrolyte, solid-state lithium battery, solvate shell, NMR, self-diffusion coefficients, chemical shifts, quantum chemical modeling

## Abstract

The effect of the composition of liquid electrolytes in the bulk and at the interface with the LiFePO_4_ cathode on the operation of a solid-state lithium battery with a nanocomposite polymer gel electrolyte based on polyethylene glycol diacrylate and SiO_2_ was studied. The self-diffusion coefficients on the 7Li, 1H, and 19F nuclei in electrolytes based on LiBF_4_ and LiTFSI salts in solvents (gamma-butyrolactone, dioxolane, dimethoxyethane) were measured by nuclear magnetic resonance (NMR) with a magnetic field gradient. Four compositions of the complex electrolyte system were studied by high-resolution NMR. The experimentally obtained ^1^H chemical shifts are compared with those theoretically calculated by quantum chemical modeling. This made it possible to suggest the solvate shell compositions that facilitate the rapid transfer of the Li^+^ cation at the nanocomposite electrolyte/LiFePO_4_ interface and ensure the stable operation of a solid-state lithium battery.

## 1. Introduction

The preparation of solid-state batteries is one of the solutions to the safety problems of lithium and lithium-ion batteries (LIBs) [1,2,3,4,5,6,7,8,9]. However, in solid-state devices, each stage of charge transfer, from the diffusion of ions in the electrolyte and electrode materials to the charge transfer across the porous electrode-electrolyte interface, is significantly hindered. At the same time, the lithium metal/polymer electrolyte interface does not form a solid electrolyte interface (SEI) on the lithium surface due to the absence of a liquid organic phase [10,11,12,13].

To eliminate the problem of poor electrode/solid electrolyte contact, we used “Liquid phase therapy” [14,15,16,17,18,19,20,21,22]. “Liquid-phase therapy” consists of the introduction of liquid electrolytes to increase ion transport and stability at the interface. The process of ion transfer is carried out only in a solid-state electrolyte. This approach is used for systems where porous structures such as a carbon anode, LiFePO_4_ or LiCoO_2_ cathodes, and solid inorganic electrolytes or their hybrids with polymers act as electrode materials [23,24,25,26,27]. Here, poor contact between solid electrolytes and electrodes is caused by their different surface morphology.

The mechanism of “liquid-phase therapy” consists of several stages:

(1) first, the lithium ion must pass from a solid (polymer, ceramic, or composite) electrolyte into a solvate shell of solvent molecules;

(2) then the solvated lithium ions are adsorbed on the electrode surface;

(3) then their partial desolvation and migration over the cathode surface with the remaining solvent molecules occur;

(4) and, finally, the detachment of the last solvent molecules and the introduction of a lithium ion into the electrode lattice take place.

In total, taking into account “liquid-phase therapy”, several interface boundaries can be observed in lithium solid-state batteries [28,29,30,31,32,33]: (1) solid electrolyte/lithium anode, (2) solid electrolyte/cathode, (3) liquid electrolyte/cathode, and (4) solid electrolyte/liquid electrolyte.

Solid-to-solid interfacial contact for the first boundary (with lithium) leads to a slow migration of ions. At the same time, for the polymer electrolyte/lithium contact, a minor number of dendrites are formed on the anode surface as was already mentioned above [10,11,12,13]. In the case of the lithium interface with nanocomposite polymeric gel electrolytes [34,35,36,37,38,39,40], the charge transfer resistance decreases for two reasons. First, the polymer electrolyte–lithium contact does not lead to the growth of SEI of a large thickness as occurs in the liquid phase. Second, oxide nanoparticles (SiO_2_, TiO_2_, Al_2_O_3_, etc.) are able to form nanostructures at the interface with lithium and to form favorable pathways for the transport of lithium ions. Therefore, in the case of using NPE, “liquid-phase therapy” at the border with lithium is superfluous.

At the solid electrolyte/cathode interface [41], it is already necessary to use “liquid-phase therapy” because of lacking tight contact. In the future, a new interface boundary the liquid electrolyte/cathode appears. Here, the transfer resistance of Li^+^ ions mainly depends on the cathode. The structure of the cathode during the charge-discharge process also greatly affects the intercalation and extraction of Li^+^ ions.

Thus, the preparation of a solid-state lithium battery with a nanocomposite gel electrolyte looks very promising. Polymer nanocomposite electrolytes (NPEs) [42,43,44] are among the promising classes of polymer electrolytes. They combine the advantages of gel electrolytes (high conductivity in the liquid phase) and composite electrolytes (good mechanical properties and a wide window of electrochemical stability).

In [45,46,47,48], we developed and studied a number of new NPEs based on a three-dimensional network matrix with ethylene oxide units, a large amount of liquid electrolyte (about 80 wt.%), and SiO_2_ nanoparticles (Aerosil 380). These nanoparticles, as shown in [48,49,50], have the property of increasing the number of charge carriers due to ionic dissociation on their developed surface.

NPE based on 15 wt.% polyethylene glycol diacrylate, 1 M LiBF_4_ in gamma-butyrolactone with 2–10 wt.% SiO_2_ nanoparticles had attractive properties, namely, high conductivity of 1–3 mS cm^−1^ at temperatures from −70 to +100 °C, high self-diffusion coefficients on ^7^Li (1.2 × 10^−10^ m^2^ s^−1^), and transfer numbers for the lithium cation up to 0.49 [45,46,47,48].

In [51], the composition of this NPE for operation in a Li/LiFePO_4_ solid-state battery was optimized. A way to reduce the resistance at the NPE/LiFePO_4_ interface was found experimentally. A high discharge capacity of ~170 mAh g^−1^ at C/10 (18 mA g^−1^) by modifying the LiFePO_4_ cathode surface with a liquid electrolyte of the 1 M LiTFSI in dioxolane/dimethoxyethane (1:1 *v*/*v*) was achieved. However, the nature of this effect remains unclear. It was more logical to assume that the composition of the liquid electrolyte for wetting the interface with LiFePO_4_ should be similar to that included in the NPE. Here it turned out, on the contrary, that when wetting with an electrolyte of a similar composition, the interface “does not work”, and when wetting with a completely different electrolyte (both salt and solvent change), the interface “opens”. A change in the composition of the electrolyte leads to a change in the solvate environment of Li^+^ ions, which takes part in electrode reactions [52,53,54].

In order to reveal possible reasons for the observed effect, in this work the influence of the solvate environment of lithium cations on the resistance of the NPE/LiFePO_4_–cathode interface was studied. In addition, high-resolution NMR and pulsed magnetic field gradient (PGM) NMR techniques in combination with quantum chemical modeling of solvate complexes in various solvents were applied.

The NMR method [55,56] and theoretical research methods (quantum chemistry, molecular dynamics) [57,58,59,60] were used to understand the state of Li^+^ ions in solution, since they are very informative for studying the mechanisms of processes occurring in LIBs.

## 2. Materials and Methods

### 2.1. Materials

LiBF_4_ (purity 98%); lithium bis(trifluoromethanesulfonyl)imide, and LiTFSI (purity 99%, water ≤ 1%) were used as electrolyte salts; gamma-Butyrolactone, (GBL, purity > 99%, water < 0.005%), 1,3-dioxolane (DOL, purity 99.8%) and dimethoxyethane (DME, purity 99%, water < 0.005%) were used as electrolyte solvents. All chemical reagents and diluents were acquired from Sigma-Aldrich (St. Louis, MO, USA) and used as received. N-Methylpyrrolidone (NMP, Fluka) was used to prepare the cathode mass. Polyethylene glycol diacrylate (PEG-DA, Aldrich, M_n_ = 575, *T*_g_ = −73.5 °C) was used to obtain a three-dimensional network matrix for the polymer electrolyte. The radical polymerization initiator, benzoyl peroxide (PB, Aldrich), stored in water (30%) was recrystallized from chloroform followed by drying at 20 °C in air and then in a vacuum. SiO_2_ nanoparticles (Aerosil 380, surface area380 m^2^ g^−1^, average particle size 7 nm) were used to fill the electrolyte polymer matrix. Lithium foil (JSC “Lithium-element”, Saratov, Russia) 1 mm thick was the anode material. The cathode components were as follows:(1) LiFePO_4_ (MTI Corporation, Richmond, CA, USA), purity >97%, particle size less than 5 µm, (2) conductive carbon black Timical Super C65 (MTI Corporation, USA, S_sp_ = 80 m^2^ g^−1^, particle size 60 µm, and (3) polyvinylidene difluoride (PVDF) polymer binder (Kynarflex HSV 900, Arkema, Colombes, France, MM > 100,000, density 1.76 g cm^−3^).

### 2.2. Liquid Electrolytes

The compositions of the electrolytes prepared for the study are given in Table 1.

The conductivity of an organic electrolyte was measured by conductometry on an LCR819 immittance meter (Goodwill Instruments Ltd., Taiwan) at an alternating current of 1 kHz in a glass electrochemical cell with plate-like platinum electrodes.

### 2.3. Synthesis of Nanocomposite Polymer Electrolyte

The nanocomposite polymer electrolyte was synthesized by the radical polymerization of PEG-DA in the presence of the radical initiator PB.

PEG-DA



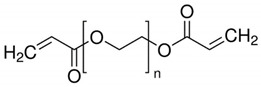



The composition of the polymerizable mixture (wt.%) was as follows: 15 PEG-DA, 78 [1 M LiBF_4_ in GBL], 6 SiO_2_, 1 PB. The curing of this mixture was carried out on the basis of studying the kinetics of polymerization [45] according to the following regime: 60 °C; for 3 h, 70 °C for 1 h, 80 °C for 1 h, and 120 °C for 1 h [51].

### 2.4. Electrode Preparation and Cell Assembly

Lithium as an anode in the form of a disk 16 mm in diameter and 1 mm thick was used. The cathodes were prepared from three components, namely, LiFePO_4_: carbon black: PVDF = 75:20:5 wt.%. PVDF was dissolved in NMP at the ratio of 2.5 mL of the solvent per 1 g of the cathode material with magnetic stirring at 50 °C. Then a weighed amount of conductive carbon black and LiFePO_4_ was added. The prepared mixture was applied onto graphitized Al foil using the Doctor Blade method (Dongguan city, China, Gelon) and then it was dried at 150 °C for 4 h. Next, the cathodes were pressed on rollers, and then it was dried for 8 h at a temperature of 120 °C in a vacuum oven.

Coin—CR2032 cells were assembled in an MBraun argon box (Germany). Symmetrical LiFePO_4_//LiFePO_4_ cells for studies by the electrochemical impedance method were used. Full Li//LiFePO_4_ cells for studies by the galvanostatic cycling method were used. An NPE membrane of the same diameter was placed between the cathode and anode.

### 2.5. Cell Testing

Electrochemical impedance measurements in symmetrical LiFePO_4_//LiFePO_4_ cells using a Z-2000 Elins impedance meter (Russia) (frequency interval 1 Hz to 600 kHz) with a signal amplitude of 10 mV were performed. Impedances are measured one week after assembling at ambient temperature.

The Li//LiFePO_4_ cells were tested on a BTS 5V10mA device (Shenzhen Neware electronic Co., Ltd., Shenzhen, China) in the galvanostatic mode at a current rate of C/10 in a range of 2.6–3.8 V.

### 2.6. NMR with Pulsed Magnetic Field Gradient

The NMR measurements on a Bruker Avance-III 400 MHz NMR spectrometer equipped with the diff60 gradient unit (the maximum field gradient amplitude was 30 T/m) were carried out at the temperature 22 ± 1 °C. The NMR measurements of ^1^H (diffusion of solvent molecules), ^7^Li (diffusion of lithium cations), and ^19^F (diffusion of anions) were carried out with operating frequencies of 400, 155.5, and 376.5 MHz, respectively. The stimulated spin-echo sequence was applied. The details of self-diffusion coefficient measurements are given in [61,62]. The experimental NMR parameters of pulse sequences were the following: π/2 pulse was9.5 μs (^1^H), 9 μs (^7^Li), and 10 μs (^19^F); gradient pulse duration time δ was 1–1.5 ms; and the diffusion 32 steps with maximum field gradient amplitude *g* were 3.5 (^1^H), 11.5 (^7^Li), and 4.0 (^19^F) T/m. The measurement error of the self-diffusion coefficients was 5%.

### 2.7. High-Resolution NMR

High-resolution spectra for ^1^H, ^7^Li, ^11^B, ^13^C, and ^19^F were recorded on a Bruker Avance III 500 MHz NMR spectrometer. The measurements at frequencies of 500, 194, 160, 126, and 471 MHz for ^1^H, ^7^Li, ^11^B, ^13^C, ^17^O, and ^19^F, respectively, were carried out at room temperature (22 ± 1 °C). Liquid samples in standard 5 mm NMR tubes without adding a deuterium solvent were placed. The chemical shift scale was calibrated with the DMSO-d_6_ signal in the capillary as an external standard (2.50 ppm for ^1^H). The ^1^H, ^7^Li, ^11^B, and ^19^FNMR spectra were obtained using the standard sequence π/2 pulses, FID. No signal accumulation was applied. To obtain the ^13^CNMR spectra, a standard sequence from the TopSpin (Bruker) zgpg30 library was used. The sequence is an accumulation of signals from 300 pulses with the suppression of the ^1^H spin-spin interaction for the duration of all the experimental times. The number of repetitions was ns = 512, and the delay between the repetition sequence was d1 = 1.5 s.

### 2.8. Quantum Chemical Modeling

The structure of solvate complexes of lithium cations with solvent molecules was studied using the nonempirical Perdew–Burke–Erzernhof (PBE) exchange-correlation functional [63] using the extended basis H [5s1p/2s1p], C, N, O, F, S [5s5p2d/3s3p2d], Li [4s1p/2s1p] for valence electrons and SBK pseudopotential [64]. The geometry of larger systems containing a counterion and additional solvent molecules was optimized using the effective Hamiltonian method [65] taking into account the van der Waals interaction. Chemical shifts for optimized structures were calculated using the Λ2 basis set of cc-pVTZ quality [66]. The Priroda package [67] was used for all the calculations carried out at the Joint Supercomputer Center of the Russian Academy of Sciences.

## 3. Results and Discussion

### 3.1. Investigation of the NPE/LiFePO_4_-Cathode Interface

At the first stage, the NPE/LiFePO_4_ interface was studied by the method of electrochemical impedance in symmetrical LiFePO_4_//LiFePO_4_ cells. The coin-type cells were assembled with an NPE membrane and various LiFePO_4_ cathode surface treatments. The cell compositions are shown in Appendix A, ESI. The Nyquist plots of the studied cells are shown in Figure 1.

The equivalent circuits for the Nyquist plots of 1–3 cells (Figure 1) were selected (Appendix A, ESI). Appendix A, ESI presents the results of calculating the parameters of equivalent circuits using the ZView2 software.

The electrolyte LiBF_4_–GBL for the preparation of NPE was chosen., The LiBF_4_ salt is thermally stable and suitable for the synthesis of polymer electrolytes by radical polymerization in contrast to the LiPF_6_ salt, which is prone to decomposition with the formation of PF_5_ [68]. This Lewis acid is the initiator of a parallel process of ionic polymerization, which ultimately leads to the deterioration of the conductive properties of polymer electrolytes [69].

The equivalent circuits of the Nyquist plots of the cells with NPE and NPE/LiTFSI-DOL-DME contain a closed Warburg element (Appendix A, ESI). That is, at low frequencies, a large number of ions that have reached the opposite electrode can intercalate into LiFePO_4_.

Figure 1 and Appendix A, ESI show the resistance at the electrolyte/LiFePO_4_ interface, which is minimum for composition No. 3. Therefore, the NPE and surface treatment of LiFePO_4_ cathodes with the LiTFSI-DOL-DME electrolyte were chosen for testing in battery prototypes.

### 3.2. Li/NPE/LiFePO_4_ Cells Cycle Tests

Based on the electrochemical impedance data, two types of Li/NPE/LiFePO_4_ cells were assembled and are shown in Figure 2, where LE is the liquid electrolyte.

Figure 3 shows the charge-discharge profiles of Li//LiFePO_4_ cells of both types. Cycling was carried out in the voltage range of 2.6–3.8 V since a 1 M solution of LiTFSI in DOL/DME (1:1) has an electrochemical stability up to 3.8 V vs. Li^+^/Li [70].

Figure 3 shows that, when treated with a liquid electrolyte, the distance between the charge and discharge plateaus decreases to 0.09 V compared to the “dry assembly”, where this value is 0.25 V. The larger voltage separation is due to a higher resistance at the electrode/electrolyte interface.

Figure 4 shows the dependence of the discharge capacity on the cycle number of the Li/NPE/LiFePO_4_ cells for different assemblies (Figure 2).

### 3.3. The Self-Diffusion Coefficients. Pulse Field Gradient of NMR (PFG NMR)

The self-diffusion coefficients D_s_ were measured by PFG NMR using the stimulated spin echo pulse train method. The dependences of the spin echo damping *A*(2*τ*_1_, *τ*_2_, *g*) on the squared amplitude of the magnetic field gradient *g^2^* (diffusion decay) were analyzed. The diffusion decay was exponential. The diffusion decays were approximated by the exponent.
(1)$$A\left(2\tau_{1},\tau_{2},g\right)=\exp\left(-\gamma^{2}\cdot g^{2}\cdot \delta^{2}\cdot t_{d}\cdot D_{s}\right)$$
where *D_s_* is the self-diffusion coefficient, *A*(2*τ*_1_, *τ*_2_, *g*) is the intensity of the “spin echo” signal, *g* is the magnetic field gradient amplitude, *γ* is the gyromagnetic ratio of the nucleus under study, *δ* is the gradient pulse duration, and *t_d_* is the diffusion time.

Figure 5 shows the diffusion decay on ^1^H, ^19^F, and ^7^Li nuclei. Table 2 presents the results of measuring the diffusion coefficients.

Diffusion attenuations on ^1^H were recorded from the fading of signals of the solvent. Depending on the composition of the electrolyte, signals from different solvents were used: GBL, DOL, and DME (Figure 5a). Thus, the ^1^H diffusion coefficients (D_s_) correspond to the mobility of the solvent molecules GBL, DOL, and DME, depending on the signal selected for diffusion decay.

Only one signal is observed in the ^7^Li NMR spectra in the salts mixture (Figure 5b). Therefore, it is impossible to obtain the diffusion coefficients D_s_ of Li^+^ cations for LiBF_4_ and LiTFSI separately. For the composition LiBF_4_-LiTFSI-GBL/DOL/DME, the measured diffusion coefficient on ^7^Li corresponds to the average value of the mobility of Li^+^ in LiBF_4_ and LiTFSI (Figure 5b).

An analysis of diffusion experiments on ^19^F (Figure 5c) allowed us to determine the average diffusion coefficient of the solvated anion BF_4_^−^ and the ion pair Li^+^BF_4_^−^ (diffusion attenuation of the ^19^F signal is −155 ppm). The average diffusion coefficient of the solvated anion TFSI^−^ and the ion pair Li^+^TFSI^−^ (diffusion signal attenuation ^19^F is −80 ppm) was also determined from the graph (Figure 5c).

As can be seen from the data obtained, the Li^+^ cations and BF_4_^−^ anions in the GBL solvent have the lowest diffusion mobility. This is probably due to the highest solvation ability of GBL among all solvents used. GBL molecules have the lowest mobility: D_s_= 4.0 × 10^−10^ m^2^ s^−1^ (composition 1) vs. D_s_ for DOL and DME molecules, 1.4 × 10^−9^ m^2^ s^−1^ and 1.0 × 10^−9^ m^2^ s^−1^ correspondingly, in average for compositions 2 and 3. However, in the case of a mixture of all three types of solvents (composition 4), their mobility becomes more comparable 4 × 10^−10^−9 × 10^−10^ m^2^ s^−1^. There is the highest decrease in DME self-diffusion coefficients in comparison with their values for DOL and GBL when changing from pure solvents to electrolyte solutions. Probably this is because of their ability to form chelate complexes (See Figure 6).

In the case of LiTFSI, the mobility of the cation and anion changes are comparable for compositions 3 and 4. At the same time, for LiBF_4_, the change to DOL/DME solution (composition 2) leads to a significant increase (by a factor of 3–4) in both cationic and anionic mobility with respect to composition 1. This is not only the effect of significantly higher mobility of DOL and DME molecules, which is higher than the mobility of GBL molecules but is associated with the formation of more mobile ion pairs as well.

The self-diffusion coefficients of solvents were determined individually (row 5 of Table 2) and in the DOL/DME mixture (row 6 of Table 2). DOL and DME molecules have the highest diffusion coefficients, and their mobility slightly decreases when a mixed solution of DOL/DME (1:1) is formed.

In the electrolytes the mobility of DOL and DME molecules of solvents is reduced two or more times due to their involvement in the coordination sphere of less mobile ions and the effect is less pronounced for LiBF_4_ solution (composition 2). In this case, the conductivity is the lowest due to the excess formation of neutral ion pairs of Li^+^BF_4_^–^. Apparently, they have less strongly bound solvate shells, in contrast to ions. Accordingly, the proportion of free solvent molecules increases, which is accompanied by an increase in their diffusion coefficients (for DOL by 1.45 times, for DME by 1.78 times) in comparison with LiTFSI solution in DOL/DME with the highest conductivity. This is comparable to the increase in the mobility of ^7^Li by a factor of 1.45 and that of ^19^F by a factor of 1.75.

The self-diffusion coefficients of DOL and DME in composition 2 are approximately equal to the average of their values in composition 3 and a pure DOL/DME mixture. This is consistent with the assumption about the formation of ion pairs and the release of pure solvent in the case of composition 2.

There is some enhancement of GBL molecule mobility in composition 4 in comparison with pure GBL solvent; this may be associated with more strong pair interactions of GBL molecules with high dipole moments. In this GBL/DOL/DME mixed solvent, one should expect a decrease in the proportion of GBL associates due to entropy effects in dilute GBL solution and a corresponding increase in the content of more mobile non-associated GBL molecules.

### 3.4. High-Resolution NMR

Table 3 and Table 4 show the ^1^H and ^13^C chemical shifts for the solvents, respectively. Appendix A, ESI show the ^1^H and ^13^C spectra of pure solvents and their mixtures along with their assignment to specific atoms of the structures.

Table 5 shows the ^7^Li, ^11^B, and ^19^F chemical shifts of the electrolyte samples.

Appendix A, ESI show the ^1^H, ^7^Li, ^11^B, ^19^F, and ^13^C NMR spectra along with their assignment to specific atoms of the structures.

### 3.5. Results of Quantum Chemical Modeling

The simplest molecular models containing one cation, one anion, and 12 solvent molecules (12 GBL molecules or 7 DOL molecules and 5 DME molecules) were used to study the structure of solvate complexes in various solvents. This stoichiometry corresponds to the composition of a 1 M solution. The van der Waals interaction of molecules in a liquid plays an important role and, therefore, it should be taken into account for a correct description of solvate complexes. For this purpose, the effective Hamiltonian method [65] was used. This technique takes into account the contribution of van der Waals interactions and is not inferior in accuracy to more time-consuming calculations by the density functional method [71]. Two types of structures containing a contact ion pair and a cation and anion separated by solvent molecules were considered. Figure 6 shows the structures of these model solvate complexes.

For the structures obtained, the magnetic shielding constants of the nuclei were calculated. Since solvate complexes differing in the arrangement and orientation of solvent molecules have similar energies, the shielding constants were averaged over several structures. This approach gives a reasonable description of ^1^H chemical shifts for pure solvents, which are modeled by clusters of 12 molecules: GBL: 2.44, 2.58, and 4.83 ppm; DOL/DME: 3.87, 5.21 ppm/3.36, 3.57 ppm. The theoretical values of chemical shifts have an inevitable error. To reduce it, when comparing the results for two types of solvate complexes, the following technique was applied.

In order to reproduce exactly the average value for all chemical shifts of nuclei of a certain type of the selected solvent molecule (*C*) and in the maximal spread (the difference between the maximum and minimum chemical shift values) (*S*) the following correction function was chosen for pure solvents:(2)$${\tilde{\delta }}_{i}=C_{exp}+\frac{S_{exp}}{S_{theor}}\left(\delta_{i}-C_{theor}\right)$$
where ${\tilde{\delta }}_{i}\;$ is the corrected chemical shift for the *i*-th nucleus, and $\delta_{i}\;$ is the calculated chemical shift. The corrected values of ^1^H chemical shifts for different types of solvent molecules were calculated (see Appendix A, ESI) using this equation.

As a result, for a solution of LiBF_4_ in GBL, the theoretical ^1^H chemical shifts for separated ion pairs are in better agreement with the experimental ones than for contact ion pairs: the root-mean-square error is 0.17 and 0.21 ppm, respectively.

For other cases, the situation is reversed: the data for contact ion pairs somewhat better describe the experimental data for solutions of LiBF_4_ and LiTFSI in DOL/DME; the corresponding root-mean-square differences are 0.16, 0.24 and 0.18, 0.19 ppm, respectively. Thus, the results of quantum chemical modeling are consistent with the NMR data on the mobility of various ions from which a higher degree of dissociation of LiBF_4_in GBL follows. A small increase in conductivity for a LiTFSI solution in DOL/DME compared to a LiBF_4_ solution in GBL can be associated with a significant increase in the mobility of ions, which is compensated by their smaller number.

During the electrode reaction, the Li^+^ ion is transferred from the solvate complex to the electrode surface at the anode. As the final state of the process is the same for different electrolytes, the corresponding energy change is determined mainly by the formation energy of the Li^+^ solvation complex. Apparently, the transfer process is activated by the formation of a coordination vacancy in the solvate complexes. This allows one to form the first intermediate structures of Li^+^ ion transfer, which are responsible for the Li^+^ transfer to the layer surface that becomes possible in this case.

These energies of various Li^+^ solvate complexes with GBL and DOL/DME molecules were found using the PBE density functional method (see Appendix A, ESI). The main conclusions are the following. First, the solvate complexes in GBL are more stable compared to the solvate complexes of DOL/DME. Second, higher energy is required for the formation of a vacancy in the coordination sphere by the removal of one of the solvent molecules: 14 and 3–9 kcal/mol, respectively. From these data, the effect of “liquid-phase therapy” can be related to a lower reaction overvoltage at the cathode due to the higher energetic availability of lithium ions in the DOL/DME solvent.

## 4. Conclusions

The resistance of the interface between an electrode based on LiFePO_4_ and a nanocomposite polymer gel electrolyte (NPE) obtained by the polymerization of diacrylate polyethylene glycol in a 1 M solution of LiBF_4_ in gamma-butyrolactone (GBL) with the addition of nanodispersed SiO_2_ was measured by a.c. electrochemical impedance spectroscopy. It was found that the NPE/cathode interface has a high resistance, while the use of “liquid-phase therapy” by 1 M LiTFSI in a mixture of dioxolane/dimethoxyethane (DOL/DME) significantly reduces this resistance. The cycling efficiency of the Li/NPE/LiFePO_4_ cells was studied. It was found that in the presence of 1 M LiTFSI in DOL/DME at the interface with the cathode, both the discharge capacity and the cycle performance significantly increase.

The self-diffusion coefficients of solvent molecules, as well as the Li+ cation and anions in solutions of 1M LiBF_4_ in GBL and DOL/DME, 1M LiTFSI in DOL/DME, and also in a mixture of 1M LiBF_4_ in GBL + 1M LiTFSI in DOL/DME, which models the composition at the NPE/liquid electrolyte interface, were measured. From the data obtained, it follows that a higher rate of salt dissociation is found in GBL, and higher ion mobility is found in a DOL/DME mixture.

It follows from the performed quantum chemical calculations that the effect of “liquid-phase therapy” at the NPE/1M LiTFSI interface in DOL/DME is due to the formation of more labile Li^+^ solvate complexes in DOL/DME, which facilitates the transfer of Li+ from NPE through a liquid solution to the solid layer of the LiFePO_4_ electrode.

## Figures and Tables

**Figure 1 membranes-12-01111-f001:**
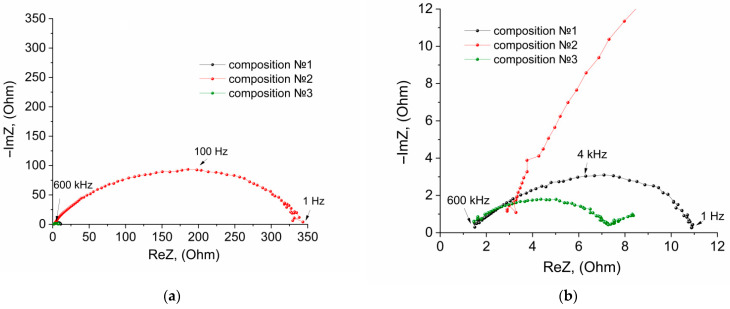
Nyquist plots of the impedance of the LiFePO_4_//LiFePO_4_ cells, where (**a**) general view and (**b**) view on an enlarged scale. Composition: (1) LiFePO_4_/NPE/LiFePO_4_; (2) LiFePO_4_/NPE/LiBF_4—_GBL/LiFePO_4_; (3) LiFePO_4_/NPE/LiTFSI—DOL-DME/LiFePO_4_.

**Figure 2 membranes-12-01111-f002:**
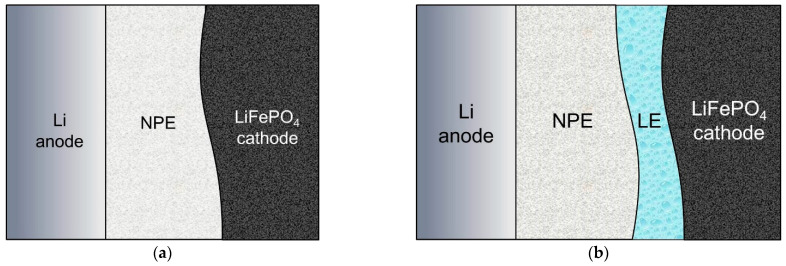
Assembly diagram of the Li/NPE/LiFePO_4_ cells, where (**a**) LiFePO_4_ surface was not treated in any way (“dry assembly”); (**b**) the surface of the LiFePO_4_ cathode was treated with a LiTFSI solution in DOL-DME.

**Figure 3 membranes-12-01111-f003:**
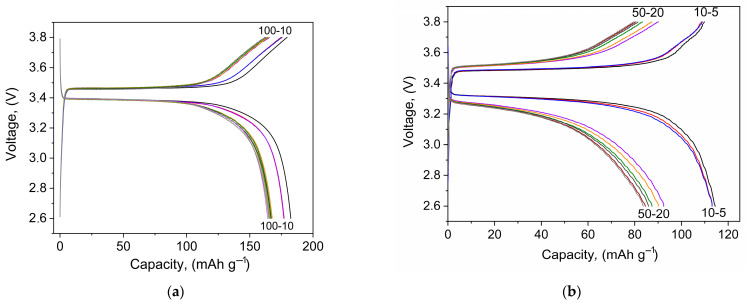
Charge and discharge curves of the Li//LiFePO_4_cells, where (**a**) with LiTFSI–DOL-DME and (**b**) “dry assembly” at the C/10 current rate in a voltage range of 2.6–3.8 V.

**Figure 4 membranes-12-01111-f004:**
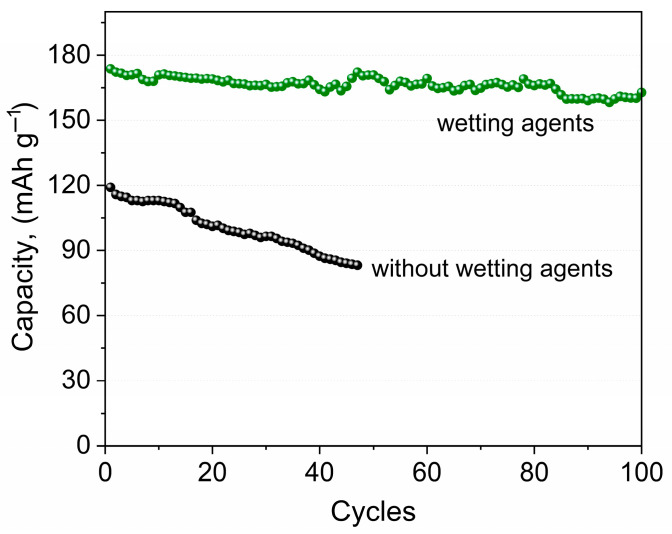
Dependence of the discharge capacity on the cycle number for the Li//LiFePO_4_ cells, NPE with liquid electrolyte 1 M LiTFSI DOL/DME (1:1) (green line) and without it (black line) at the C/10 current rate in a voltage range of 2.6–3.8 V.

**Figure 5 membranes-12-01111-f005:**
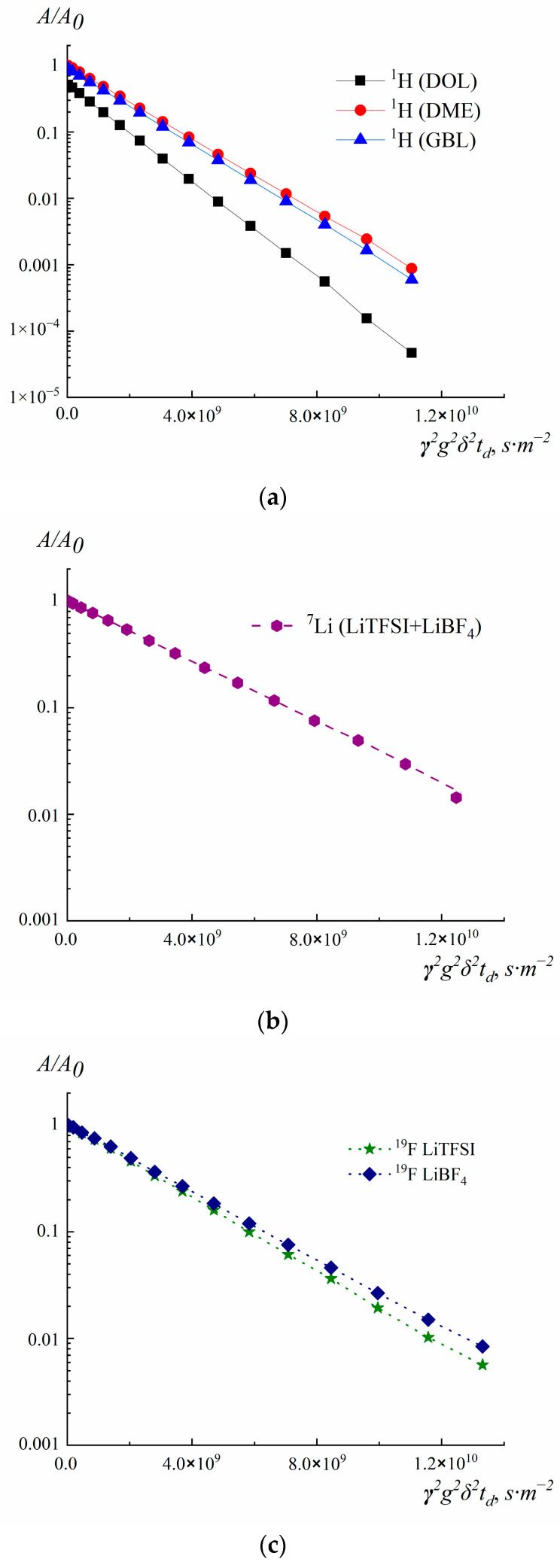
Diffusion decays on (**a**) ^1^H, (**b**) ^7^Li and (**c**) ^19^F.

**Figure 6 membranes-12-01111-f006:**
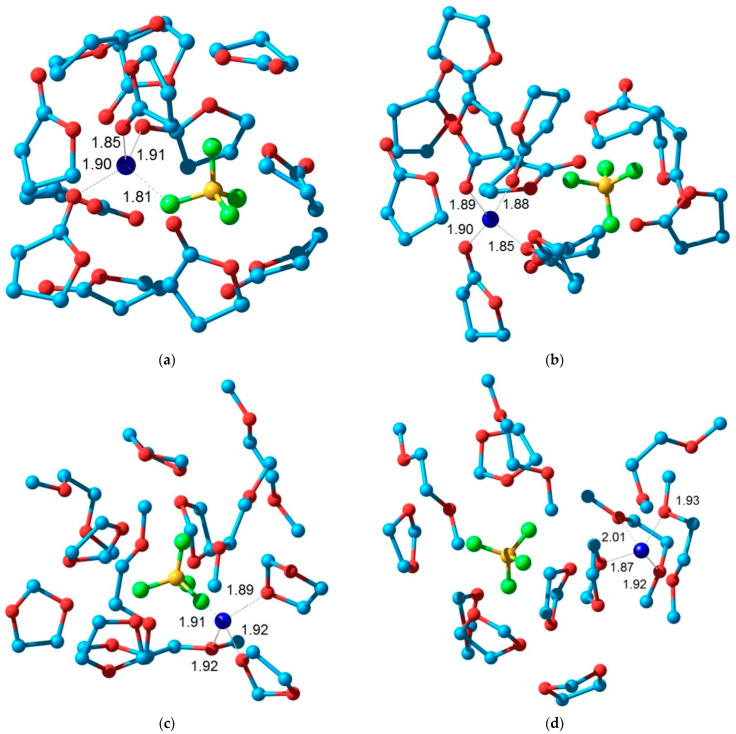
Structure of the model solvate complex including the ion pair (**a**,**c**,**e**) and separated ions (**b**,**d**,**f**) for the electrolytes: LiBF_4_ in GBL (**a**,**b**); LiBF_4_in DOL/DME (**c**,**d**) and LiTFSI in DOL/DME (**e**,**f**).

**Table 1 membranes-12-01111-t001:** Compositions of liquid electrolytes.

No.	Electrolyte	Abbreviation
1	1 M LiBF_4_ in GBL	LiBF_4_—GBL
2	1 M LiBF_4_ in DOL/DME (1:1 *v*/*v*)	LiBF_4_—DOL/DME
3	1 M LiTFSI in DOL/DME (1:1 *v*/*v*)	LiTFSI—DOL/DME
4	1 M LiBF_4_ in GBL + 1M LiTFSI in DOL/DME (1:1 *v*/*v*)	LiBF_4_—LiTFSI—GBL/DOL/DME

**Table 2 membranes-12-01111-t002:** Self-diffusion coefficients on ^7^Li, ^19^F, and ^1^H (m^2^ s^−1^) and experimental conductivity at room temperature.

Sample	LiBF_4_ (^7^Li)	LiTFSI (^7^Li)	LiBF_4_ (^19^F)	LiTFSI (^19^F)	GBL (^1^H)	DOL (^1^H)	DME (^1^H)	Conductivity, mS cm^−1^
**(1) LiBF_4_—GBL**	1.8 × 10^−10^		2.5 × 10^−10^		4.0 × 10^−10^			8.0
**(2) LiBF_4_—DOL/DME**	6.8 × 10^−10^		7.0 × 10^−10^			1.6 × 10^−9^	1.3 × 10^−9^	5.7
**(3) LiTFSI—DOL/DME**		4.7 × 10^−10^		4.0 × 10^−10^		1.1 × 10^−9^	7.3 × 10^−10^	9.4
**(4) LiBF_4_—LiTFSI—GBL/DOL/DME**	3.3 × 10^−10^		3.6 × 10^−10^	3.9 × 10^−10^	6.4 × 10^−10^	8.5 × 10^−10^	6.5 × 10^−10^	10
**(5) Solvents**					7.2 × 10^−10^	2.1 × 10^−9^	2.8 × 10^−9^	
**(6) DOL/DME(1:1)**						2.0 × 10^−9^	2.2 × 10^−9^	

**Table 3 membranes-12-01111-t003:** Experimental ^1^H chemical shifts of the solvent and electrolyte samples (in ppm).

Solvents				Electrolytes			
GBL	DOL	DME	DOL + DME	No. 1	No. 2	No. 3	No. 4
4.09				4.08			4.14
2.23				2.22			2.27
2.01				1.98			2.03
	4.58		4.72		4.67	4.66	4.59
	3.57		3.71		3.68	3.67	3.61
		3.61	3.38		3.38	3.40	3.31
		3.45	3.22		3.21	3.22	3.12

**Table 4 membranes-12-01111-t004:** Experimental ^13^C chemical shifts of the solvent and electrolyte samples (in ppm).

Solvents				Electrolytes			
GBL	DOL	DME	DOL + DME	No. 1	No. 2	No. 3	No. 4
177.70				179.00			179.03
68.19				68.69			68.83
26.91				27.05			27.27
21.54				21.37			21.62
	94.08		94.22		94.08	94.08	94.30
	63.66		63.80		63.77	63.75	64.03
		71.67	71.35		70.80	70.72	70.97
		57.84	57.55		57.61	57.71	57.87

**Table 5 membranes-12-01111-t005:** Experimental chemical shifts of the electrolyte samples (in ppm).

Electrolytes	^7^Li	^11^B	^19^F		
No. 1	−0.45	−1.47	−154.94	−154.99	
No. 2	−1.45	−1.37	−155.54	−155.60	
No. 3	−1.69	-	−80.19		
No. 4	−0.87	−1.43	−80.06	−155.18	−155.24

## Data Availability

Not applicable.

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
