# Peer review of "Effect of the Solvate Environment of Lithium Cations on the Resistance of the Polymer Electrolyte/Electrode Interface in a Solid-State Lithium Battery"

_membranes, 2022, doi:10.3390/membranes12111111_

Round 1
Reviewer 1 Report
The introduction was badly written. Could you more elaborate on the concept of ``liquid phase therapy`` There is a lack of references and state-of-the-art of gel-based batteries? Please also provide the loading of the cathode and the temperature of operation. LiBF4 is well known to be very reactive with Li, and now LiFSI is more impelmentant with Li metal anode. I believe testing LiBF4 as electrolytes was not useful. PFG NMR experiments are well conducted. Also, please add more details at the conclusion.
Author Response
Thanks for the good review. Thanks to your comments, the text has been revised and improved.

Reviewer 2 Report
Revision of membranes-1971041
There is attached an annotated pdf file with some formal comments on the MS.
My major comments are the following:
Section 3.3; the differences/changes in values obtained for the different species should be discussed in terms of coordination chemistry (solvation). For example, the reduced mobility of GBL in sol. 1 compared to neat GBL is most likely a result of coordination of some GBL to Li+ as per the modelling results. A significant reduction in solvent mobility is also observed for sol. 4. There is also a mistake in the citation of tabulated diffusion data in line 256.
Statements such as “The solvent molecules DOL, DME are more mobile in the composition with LiBF4 (composition 2) than with LiTFSI (com-270 position 3) due to the larger size of LiTFSI compared to LiBF4” in line 270, p. 8 are not quite true – the mobility of DOL in sol. 3 is almost the same as in sol. 2 and the most striking difference in mobilities here is the one of DME and the differences are more likely to be related to changes in the coordination environment of Li+.
Unfortunately, a LiBF4/Dol/DME mixture was not modelled as a comparison to LiTFSI/DoL/DME which would have helped the interpretation of obtained data.
More generally in this context, the manuscript essentially presents a collection of data generated by various methods, however, there is an obvious lack of bringing these together in a more profound discussion/interpretation. I therefore ask for a major revision before this work can be published.

Author Response

(The authors gave the same response as above.)

Round 2
Reviewer 1 Report
The authors adressed probably all the previous comments. The introduction was improved.
Reviewer 2 Report
I thank the authors for addressing my comments.